



# Development of common socio-economic scenarios for climate change impact assessments in Japan

Sayaka Yoshikawa[1], Kiyoshi Takahashi[2], Wenchao Wu[3], Keisuke Matsuhashi[2], Nobuo Mimura[1]

[1] Global and Local Environment Co-creation Institute (GLEC), Ibaraki University, Japan
[2] Social Systems Division, National Institute for Environmental Studies (NIES), Japan
[3] Social Sciences Division, Japan International Research Center for Agricultural Sciences (JIRCAS), Japan

*Correspondence to*: Sayaka Yoshikawa (sayaka.yoshikawa.aa@vc.ibaraki.ac.jp)

**Abstract.** Climate change is one of the greatest long-term challenges faced by humanity. In the projection of climate change
impacts, scenarios based on assumptions regarding future conditions are commonly used. Shared socio-economic pathways
(SSPs) are widely employed as socio-economic scenarios for global-scale predictions. The SSPs provide future projections
of population and gross domestic products. However, SSPs are not suitable for detailed assessments for a country such as
Japan, as they include only global regional data. The S-18 project aims at a nationally unified projection of climate change
impacts across multiple sectors in Japan. In contribution to this, based on the previous study for Japan SSPs, we established
common socio-economic scenarios designated as Japan SSP1, Japan SSP5, and status quo. Japan SSP1 and Japan SSP5 are
based on qualitative links to global SSPs. Japan SSP1 foresees sustainable society with low−carbon emission, while Japan
SSP5 envisions a society dependent on fossil fuels, emitting large amounts of greenhouse gases. The status-quo scenario
assumes no future change based on the current conditions in Japan. Moreover, we provided a common dataset of population
and land-use under these scenarios. Population data were obtained from existing population projections, and land-use data
were estimated according to population changes and current land-use classifications. Here, the dataset prepared for the S-18
project is detailed and possibilities for its improvement discussed.

## 1 Introduction

The IPCC (2021) states that human influence has warmed the atmosphere, ocean, and land. Human-induced climate change
has already affected changes in extremes weather events, such as heatwaves, heavy precipitation, droughts, and tropical
cyclones in every region across the globe. Recently in Japan, the amount of damage caused by natural disasters has increased.
In 2019, this amount accounted for more than 20% of total damage worldwide (Guha-Sapir et al., 2022). As a country
surrounded by seas, much of the damage is caused by storms and floods due to tropical cyclones and heavy precipitation.
The number of municipalities where the Disaster Relief Act was applied due to storm and flood damage increased between
2018–2020 (Figure 1). Recent weather extremes in Japan, such as the heavy rain events in July 2018 (Imada et al. 2019;



Kawase et al. 2020) and extremely heavy precipitation induced by Typhoon Hagibis in 2019 (Kawase et al. 2021) are likely to have a robust relationship with global climate change.

As impacts of global climate change emerge, the need to adapt to their adverse effects has become apparent in Japan. Consequently, the Climate Change Adaptation Act (2018) has been in effect in Japan since Dec 2018. In 2025, the Japanese government will conduct an extensive review of national impact assessments and adaptation plans under the Climate Change
Adaptation Act. To generate the scientific information necessary for this review, a 5-year research project, called the S-18 research project, was launched by the Environmental Restoration and Conservation Agency under the Ministry of the Environment in 2020. The S-18 research project (details in Section 2) aims to implement a nationally unified projection of climate change impact and adaptation assessments for multiple sectors.

Climate change issues operate on time scales up to a century-long. Studies on climate change impacts and adaptation
measures typically use scenarios that assume some form of future conditions. Scenarios are the input data for any model, which assumes a possible future. Moss et al. (2010) proposed a new process for creating plausible scenarios to investigate climate change confronting the global community. The scenarios are sets of global change scenarios consisting of the radiative forcing (i.e., greenhouse gas (GHG) emission) scenario of the Representative Concentration Pathways (RCPs; van Vuuren et al., 2011), climate scenarios of the Coupled Model Intercomparison Project Phase 5 (CMIP5; Taylor et al., 2012),
and socio-economic scenarios of the shared socio-economic pathways (SSPs; Kriegler et al., 2012; O'Neill et al., 2012; O'Neill et al., 2017). The RCPs assume the levels of global warming toward the end of 21$^{st}$ century, which are used to estimate changes in climatic hazards, while the SSPs represents the exposure and vulnerability assuming the development pathways of society. they were the basis for climate model projections in CMIP5 and their assessment in the Intergovernmental Panel on Climate Change Fifth Assessment Report (IPCC, 2013; 2014). However, this set covers a global
scale, and therefore details required for national-scale assessments are missing.

SSPs require revised versions for local assessments in Japan because global narratives may lack important region-specific drivers, national policy perspectives, and the unification of data provided by the government. As part of the S-18 research project, this study aims to establish several socio-economic scenarios as inputs for projections of climate change impacts and evaluation of adaptation options. These projections, performed using these scenarios, are supposed to contribute to the
review of national impact assessments and adaptation plans. At the same time, as the scenarios presented are on a national-scale and general, they are intended to be used jointly with other specific assumptions made by each sector.

We believe that there are two main groups of readers of this study. The first group includes those who wish to use this prepared socio-economic scenario (including the S-18 research project); the second group includes readers who wish to use the prepared scenarios to develop their own regional scenarios within Japan or in different countries. We also provided
specific information on the development process, which could be useful for the second group. In the next section, the concept of the S-18 research project and the selection of qualitative narrative components in the socio-economic scenario are explained, followed by a description of the quantitative components of the socio-economic scenarios consisting of population and land-use. The section of future challenges discusses possibilities for its improvement.



## 2 Framework of the S-18 research project

The S-18 research project aims to create new scientific information on projection of climate change impacts and adaptation assessment to support climate change adaptation measures in all of Japan. The Climate Change Adaptation Act (2018) requires that the national adaptation plan be revised every five years through repeated impact assessments for Japan in response to the evolving scientific understanding of both climate change and its related impact. The outcomes of the S-18 project are expected to be used in the next impact assessment report to be released in 2025 as mentioned above. As of June

2022, 147 local governments have formulated regional climate change adaptation plans (A-PLAT, 2022). There are more than 1,700 local governments in Japan, many of which are still in the planning stages for their adaptation plans. For appropriate local planning, there are local governments' needs for specific information on impact projections in each municipality and effective adaptation measures. In the S-18 project, meeting these needs is also set as one of the goals.

To achieve these goals, the S-18 project has formed research teams under five overarching themes, which collaborate

closely to promote research. The five themes are as follows:

Theme 1:    Development of Comprehensive Research Framework for Impact Projection and Evaluation of Adaptation

Theme 2:    Projection of Climate Change Impacts and Evaluation of Adaptation Options for Agriculture, Forestry, and Fisheries

Theme 3:    Projection of Climate Change Impacts and Evaluation of Adaptation to Natural Disasters and Water Resources

Theme 4:    Projection of Climate Change Impacts on Quality of Life (QoL) of People and Their Associated Infrastructure and Local Industries and Evaluation of Adaptation Options

Theme 5:    Development of Economic Assessment Methods for Impact of Climate Change Adaptation Options

Figure 2 shows the structure of the S-18 research project. In Theme 1 the tasks are to develop the framework of the S-18 project and common scenarios (climate scenarios and socio-economic/adaptation scenarios), construct an integrated database to compile project outputs, develop statistical models for impact and adaptation assessment, and develop analysis methods for adaptation planning. In Themes 2, 3, and 4 the tasks are to project climate change impacts and assess adaptation

measures for their respective sectors. In Theme 5 the tasks are to develop economic evaluation methods for impacts and adaptation measures in the agriculture and local economy sectors and assess the socio-economic impacts of disasters such as typhoons and floods. The S-18 research project consists of about 200 researchers belonging to 19 sub-themes. Up-to-date project information is available at https://s-18ccap.jp/en/ throughout the project.

Next, the research targets of the S-18 research project are presented. The S-18 research project is concerned with six

sectors: Agriculture/livestock raising/forestry/fisheries, Water environment/water resources, Natural disasters/coastal areas, Human health, Industrial industry/economic activity, and Life of citizens, infrastructure, and industries.



The S-18 project generates information on the projected impacts and the effectiveness of adaptation measures in these six sectors. The study area is either all of Japan or a specific region of the case study. The spatial resolution of the national evaluation is the Japan MESH3 Boundaries (approximately 1km x 1km) or a municipality-scale. The boundaries show a

mesh of connected rectangular cells, known as the "Basic Grid Square" defined by the government of Japan. The longitude and latitude heights of each cell are 45 arcsecond and 30 arcsecond, respectively, which are approximately 1 km (tertiary mesh units). This spatial resolution is a common basis for data (climate forcing data and socio-economic data) provided within the S-18 research project. The projection period covered is from 2020 to 2100 with three time slices: the near future (2030), the medium term (2050), and the long term (2090). The outcome to be generated from the five themes is expected to

cover more than 60 impact areas.

A unique feature of the S-18 project is the establishment of a common impact assessment framework to allow for comparisons across sectors. The common framework consists of sets of climate scenarios determined by combination of RCPs and multiple climate models, socio-economic scenarios, and with/without adaptation condition. All themes in the project use a common framework for their impact projection and adaptation assessment. RCPs were considered to be in the

range of 8.5 W/m² (RCP8.5) to 2.6 W/m² (RCP2.6) assuming greenhouse gas emissions correspond to the radiative forcing until 2100. RCPs except for the RCP8.5 addresses greenhouse gas mitigation measures. The S-18 research project compares the impacts of low-end (<2°C; RCP2.6) and high-end (>4°C; RCP8.5) climate change scenarios. The climate scenarios are set up to deal with both CMIP5 (Taylor et al., 2012) and CMIP6 (Eyring et al., 2016). The socio-economic scenarios comprise three cases: two types of future change and a fixed current population and land use distribution. This study is based

on an impact risk analysis framework with hazard, exposure, and vulnerability as elements. (European Commission 2011, UNISDR 2009). The establishment of common climate and socio-economic scenarios means that hazards and exposures in impact projections are aligned. However, vulnerability will be considered in each sector. In the impact projections, how to set future socio-economic conditions is always a challenge, because the projected results vary widely depending on the socio-economic scenarios. The S-18 project uses the Japan SSPs, as shown in the next section.

**3 Selection of qualitative narratives components in socio-economic scenarios**

The socio-economic scenario represents future social and economic activities. SSPs, which are socio-economic scenarios used in global climate change studies (Kriegler et al., 2012; O'Neill et al., 2012), consist of qualitative and quantitative components. Quantitative components provide projections of key elements, including national-level population growth and educational composition (Samir and Lutz, 2017), urbanization (Jiang and O'Neill, 2017), and economic growth (Cuaresma,

2017; Dellink et al., 2017; Leimbach et al., 2017). Qualitative narratives describe the evolution of aspects of society (such as economy and lifestyle, policies and institutions, technology, environment and natural resources) that are difficult to project quantitatively, provide the logic underlying those elements of scenarios that are quantifiable, and provide a basis for further elaboration of scenarios by users.





Qualitative narratives in the SSPs depict five different global situations (SSP1–5) with different socio-economic conditions.
The five SSPs are specific combinations of challenges to mitigation and adaptation (O'Neill et al. 2017). That is, the SSPs describe scenarios in which societal trends make the mitigation of, or adaptation to, climate change harder or easier. SSP1 represents a sustainable world in which the challenge of both mitigation and adaptation to climate change is easier. SSP2 represents conditions in which there are moderate challenges to mitigation and adaptation, but with significant heterogeneities across and within countries. SSP3, with its theme of international fragmentation, represents conditions in
which the challenge to both mitigate and adapt is harder. SSP4 represents conditions where there are easy challenges to mitigation but hard challenges to adaptation, with limited access to effective institutions for coping with economic or environmental stresses. SSP5 represents a situation of strong reliance on fossil fuels and a lack of global environmental concern, meaning the challenge to climate change adaptation and mitigation is difficult.

O'Neill et al (2014) proposed that revised versions of global-scale SSPs would be needed for the regional assessment, while
the SSP has the great advantage of being available in a global model. Chen et al. (2020) outlined three reasons which are failing to regional specific important drivers like an aging society, no reflecting national policy perspectives, and no being consistency with the historical statistics for each nation. Thus, it is necessary to construct new socio-economic scenarios that can be used for the assessment of climate change impacts, adaptation, and mitigation measures by national and sub-national governments in response to the SSPs, to reflect local unique situations. Few studies (e.g., Kok et al., 2019; Pedde et al.,
2019; Zandersen et al., 2019) have applied it at the national scale, or the sub-national scale. Chen et al. (2020) proposed Japan SSPs that qualitatively link to global SSPs and basic quantitative information of Japan. Based on the discussion in the workshops, they extended the description applicable to a regional scale, correcting the outline of the global SSP. Specifically, they added descriptions on industry, employment, migration, responsiveness, and diversity as key elements affecting Japan SSP. Then, they modified the description of the population to fit the Japanese government's projections. In the S-18 research
project, we decided to use the qualitative narratives in Japan SSPs provided by Chen et al. (2020), which are the only Japan SSPs that correspond to the global SSPs. In the experimental designs for climate change impact assessments, multiple combinations of the five SSPs and the RCPs with climate model ensemble members are possible. However, it is impractical to analyze all of them. O'Neil et al. (2016) proposed four high-priority scenarios as anchor experiments. The scenarios span a wide range of uncertainty in future forcing pathways important for the collaborative work of integrated assessments and
impact/adaptation/vulnerability researchers in integrated scenarios. It includes new SSP-based scenarios as continuations of RCP2.6, RCP4.5, and RCP8.5, and an additional unmitigated forcing scenario (SSP3-7.0). In accordance with the highest priority critical scenarios by O'Neill et al. (2016), we selected Japan SSP1 for RCP2.6 (2°C equivalent) and Japan SSP5 for RCP8.5 (4°C equivalent).

The common socio-economic scenarios in the S-18 research project are as follows.
150         1.      Japan  SSP1 (Sustainability) scenario
              2.      Japan  SSP5 (Fossil-Fuel Development) scenario
              3.      Status quo (No change) scenario



In addition to Japan SSP1 and Japan SSP5, a status quo scenario (i.e., no change to current situation) was established. The status quo scenario assumes the current social conditions in Japan will continue until 2100 to be able to assess the impact of climate change only. The following four combinations of RCPs and the socio-economic scenarios were then used for the first assessment in the S-18 research project:

    A)       RCP2.6－Japan SSP1 scenario

    B)       RCP8.5－Japan SSP5 scenario

    C)       RCP2.6－Status quo scenario

    D)       RCP8.5－Status quo scenario

RCP2.6－Japan SSP1 scenario: This scenario represents the low end of the range of future forcing pathways with a sustainable society. When policy change through increasing environmental awareness in societies is achieved, global warming significantly less than 2°C by 2100 is anticipated (O'Neill et al., 2016). Japan SSP1, as described by Chen et al. (2020), foresees relatively increasing numbers of people living in local and suburban areas and utilizing new technologies using local characteristics. Moreover, decreased abandoned areas of cultivation, improved food self-sufficiency rate and appropriate management of forests (including sixth industrialization of agriculture, forestry and fishery industries) will be progressing. Japan SSP1 presents low challenges to mitigation and adaptation, in line with global SSP1.

RCP8.5－Japan SSP5 scenario: This scenario represents the high end of the range of future forcing pathways with a situation of strong reliance on fossil fuels. It is the only scenario with emissions high enough to produce over 4°C global warming by 2100 (O'Neill et al., 2016). Japan SSP5, as described by Chen et al. (2020), foresees accelerated globalization, progressed development and utilization of new technologies and a significantly greater growth of manufacturing and construction industries which lead the economy as an export industry. Population and capital will concentrate more strongly in cities. Mechanized agriculture, forestry and fishery industries will develop and spread through utilizing new technologies. Japan SSP5 presents high challenges to mitigation and low challenges to adaptation, similar to global SSP5.

RCP2.6－Status quo scenario: The social conditions in Japan are assumed to remain the same as the present under the climatic changes with RCP2.6 scenario. For the impact projection and adaptation assessment in the S-18 project, it was established to assess the impact with sole scenario of climate change under RCP2.6 compared with the impact under both scenarios of climate change and socio-economic situation.

RCP8.5－Status quo scenario: The scenario assumes no change in social conditions in Japan from the current level under the climatic changes with RCP8.5 scenario. It was designed to assess the impact with the sole scenario of climate change under RCP8.5 as in the RCP2.6－Status quo scenario.

For each sector to make effective climate change impact projections, it is necessary to select the appropriate quantitative components of socio-economic scenarios for each assessment. We investigated what type of these components the S-18 project researchers required. After surveying all 19 sub-theme teams, it was found that 80% of the teams required land-use/land-cover (LULC), and 50% required population, as shown in Table 1. The need for LULC details varied among the





teams. One team needed only changes in paddy fields and croplands, whereas other teams needed more than seven type changes, such as paddy fields, other crops, residences, golf courses, and traffic areas. Then, the quantitative components were determined on two types: population distribution by the 5-year-old age group for both 5- and 1-year and LULC for every 5 years at Territory Units and both prefectural and municipal scales as the socio-economic scenarios. The common
data are described in detail in the next section.

## 4 Quantitative components of socio-economic scenario: Population

### 4.1 Population trends and projection

The 2015 Population Census database (Statistics Bureau of Japan, 2019) was selected as the basic data for Japan SSP1, SSP5, and the status quo scenario. The census is the most important statistical undertaking covering all persons and
households living in Japan and is conducted every five years. Although the most recent census was conducted in 2020, the most recent detailed data available were from 2015, and according to that census, the total population of Japan was 127.09 million (including non-Japanese residents). The characteristics of the population structure are that the annual number of births and population of children in Japan has declined. The percentage of the population over the age of 65 is the highest in the world (United Nations, 2019).

In Japan, population projections have widely been discussed for future projection of not only impact of climate change but also other assessments (e.g., Hansaki et al., 2012; 2014; NIPSSP, 2017; Etoh and Onishi, 2018; Sugiyama et al., 2019; Matsui et al., 2019; NIES 2021). Although some have no spatial distribution and others are single-year data (each feature summary in Table A1 of Appendix A), most of them are based on scenarios from the National Institute of Population and Social Security Research under the Ministry of Health, Labour and Welfare (NIPSSP, 2017). NIPSSR is an institute that
provides demographic data and initiates the 'Population Projection for Japan', which projects the overall size and age–gender breakdown of the future population. This projection, based on information from the census, vital statistics, and various national representative surveys, is widely used by national and local governments and in various fields in Japan.

Comparing NIPSSP with population projection under global SSPs, the set birth rate in global SSPs is much higher than that in the NIPSSP. Chen et al. (2020) stated that the direct use of global SSPs data does not reflect the characteristics of a region and is likely to cause greater errors, leading to irreparable errors in policy guidance. Chen et al. (2020) then selected datasets
coherent with Japan SSPs from NIPSSP (2017), which have been widely used in Japan. Using the selected dataset by Chen et al. (2020), NIES (2021) formed the population by the municipality and tertiary mesh distribution for each 5-year age group for the Japan SSPs between 2015–2100. While the total population is made to match the selected datasets from NISSP perfectly, the total population by the municipality differed from NISSP projection due to changing ratios of outflow and
inflow for each SSP scenario. The ratio of outflow and inflow were based on the degree of concentration of the population in four categories: urban area in metropolitan region, non-urban area in metropolitan region, urban area in rural region and non-urban area in rural region.





After comparing the results of the needs assessment with the details of previous studies, it was decided to use the population projection in NIES (2021) under Japan SSP1 and Japan SSP5. The total population was predicted to decrease

over time until 2100 under both Japan SSP1 and Japan SSP5. A greater decrease in population was predicted in the Japan SSP1 than in the Japan SSP5 (Figures 3). Approximately 2.7% under the Japan SSP1 and 4.7% under the Japan SSP5 municipalities of a total 1683 municipalities show population growth from 2015 to 2100. In many cases, however, rather than an increase in the birth population, the population of the over 65 age group simply increases. More than 1,600 municipalities show a population decline in both Japan SSP1 and Japan SSP5. Population concentration in metropolitan

regions in 2100 is greater in Japan SSP5 (64.2% relative to the total population) than in Japan SSP1 (49.8%).

## 4.2 Notes for using population projection data under Japan SSP1 and Japan SSP5

The following points should be considered when using the data:

- ✓ Areas with smaller populations in 2015 (the base year) were more likely to have extreme results. The population in 2100 is expected to be less than 20% of the 2015 population in several municipalities.

- ✓ The population data assume that medical and welfare services will not deteriorate from current level and will always be provided. NIPSSP's future population projections, including fertility and mortality rates, are based on historical data.

- ✓ The population data assumed that the gender and age composition within each grid would be constant from 2020 to 2100. In 2020, the composition used the distribution of 1 km mesh-based future population data from NLID (2018)

which is based on NIPSSP (2017), because the distribution data were limited.

## 5 Quantitative components of socio-economic scenario: LULC

### 5.1 LULC and the projection

LULC changes due to agricultural expansion, urbanization associated with population growth, and economic development have occurred in several parts of Japan in the decades since World War II (Himiyama 1999). As a national-scale database of

LULC in Japan, the GIS-based "Land Utilization Tertiary Mesh Data" from NLID (2016) have been developed since 1976. The data have been published every 5 to 10 years (e.g., 1977, 1988, 1992, 1998, 2007, 2009, 2014, and 2016). Land classification data have been developed focusing on human-dominated land covers, such as residential and transportation, identified based on topographical maps and vegetation areas as detected using a normalized vegetation index calculated from satellite image data, mainly Landsat, TERRA (Aster), and ALOS. NLID classifies nationwide land usage at a 100-m

resolution. There are twelve land cover types: (1) paddy field, (2) other agricultural lands, (3) forest, (4) wasteland, (5) building area, (6) roads, (7) railways, (8) others, (9) rivers and lakes, (10) seashore, (11) sea area, and (12) golf course (details in Table 2). Each record includes areas ($m^2$) of 12 land cover types in a mesh unit. We used LULC data of NLID in





2016 as the base year data for Japan SSP1, Japan SSP5, and the status quo scenario. LULC under the status quo scenario is not changed from the LULC data from 2016 to 2100.

Future projections of LULC change are valuable for developing strategies to address the climate change impact and adaptation measures in many fields, such as natural disasters, biodiversity conservation, and carbon cycle. LULC projection should be consistent with high-resolution population projection because spatially explicit population projection has played an important role in studies predicting LULC change (e.g., Bengtsson et al. 2006; Shen et al. 2008). In most previous studies on LULC projections in Japan, demographic change was thought to be the major driver of LULC change (e.g., Hanasaki et al.,

2012; 2014; Etoh and Onishi 2018; Ohashi et al., 2019; Shoyama et al., 2019; see feature summary in Table A2 of Appendix A), causing population change has to be as a proxy variable for LULC change. However, to the best of our knowledge, no studies have developed future LULC projections associated with population projections under national specific SSPs for Japan. The assumption was made that LULC change depends on changes in population distribution, and this trend will continue in the future.

**5.2 Creation of future scenarios for LULC distribution based on population data**

LULC projection data were generated following the method of Hanasaki et al. (2012; 2014), which predicts LULC change using the above assumptions. There are two main steps in the calculation of change in LULC types. First, future changes in building area were estimated based on the relationship between building area and population change during the base year 2015. Specifically, the population change rate was determined from the populations in the base year and the next year. The

building area for the target year was calculated by multiplying the population change rate by the building area in the base year. The equation used is as follows:

$$f(i,t) = \min\left(\frac{P(IP,t)}{P(IP,t_s)} \cdot f(i,t_s), A(i,t-1)\right), i \subset IP$$

where $f$ is the building area of the mesh unit; $P$ is the population at the prefecture-level; $t$ is the year; $t_s$ is the base year; $i$ is

the number of mesh units; $IP$ is the prefecture code; $A$ is total area of building areas, paddy fields, other agricultural areaa, and wastelands. Land Utilization Tertiary Mesh Data (NLID, 2016) were used for LULC in the base year. Population data for the future years under both the Japan SSP1 and Japan SSP5 scenarios for every five years from 2015 to 2100 were used to estimate the population change rate.

Second, for the decrease or increase in building areas, conversion to or from other land use types should occur. The direct

conversion from roads, railways, others, rivers and lakes, seashores, sea areas, and golf courses to building land is considered unlikely. In addition, many cases have been reported of agricultural land being converted into building areas, such as industrial and residential areas (MAFF, 2016). Therefore, if the building area increased after the building area was calculated, the increase was assumed to be converted from other four LULC types which are paddy fields, other agricultural lands, forests, and wastelands. The conversion rules are as follows.





Rule 1:  The increase in building areas is deducted equally from the four LULC types.

Rule 2:  If the increase in building areas is not deducted equally from the four LULC types, the maximum areas of a given LULC type, which has smallest area in the four LULC types, is deducted equally from the four LULC types. Then, the area of the given LULC type become zero, and the remaining area in the increase in building areas is deducted equally from the other three LULC types.

Rule 3:  If the increase in building areas exceeds the total area of the four LULC types, the areas of these four LULC types are reduced to zero.

For the decrease in building areas, the direct conversion from building areas to other LULC types excluding wastelands is considered unlikely. Then, if the building area was reduced after the building area was calculated, the reduction was added to the area of the wastelands.

The above rules apply only to areas of LULC types within the same mesh unit. In this study, we adopted the use of 'other agricultural lands', while Hanasaki et al. (2012; 2014) adopted 'others' in the four LULC types. The 'Other agricultural land'
type is cultivated cropland except for paddy fields. The 'Other' type is public facilities including athletic fields, airports, schools, and harbor areas, and vacant lots that has been built over with human engineering. This is because it is unlikely that 'others' including public facilities would be converted into building areas Seven other LULC types which are roads, railways, others, rivers and lakes, seashores, sea areas, and golf courses were excluded from the future estimation and were assumed to have no change in area in the future.

**5.3 Future scenarios for LULC distribution based on population data**

Changes in LULC types were projected under two different scenarios which are Japan SSP1 and Japan SSP5. In Japan SSP1, paddy fields, other agricultural lands, and forests are predicted to decrease over time until 2040. After 2040, the above LULC types did not change until 2100. The total area of paddy fields decreased by 0.02%, other agricultural land decreased by 0.06%, and forests decreased by 0.01% in 2040 compared to the total area in 2015 (Figures 4a, b, and c). All three LULC
types decreased to a greater degree in Japan SSP5 than in Japan SSP1. In Japan SSP5, the three LULC types are predicted to decrease over time until 2055. the total area of paddy fields decreased by 0.07%, other agricultural land decreased by 0.1%, and forests decreased by 0.02% in 2055 compared to the total area in 2015 (Figures 4a, b, and c).

The building areas were predicted to decrease until 2100 under both scenarios. The total building areas in Japan SSP1 and Japan SSP5 was, respectively, 49% and 45% lower in 2100 than in 2015 (Figure 4d). The wasteland area was predicted to
increase until 2100 under both scenarios in response to decreasing building areas. The total wasteland area in Japan SSP1 and Japan SSP5 was, respectively, 152% and 142% higher in 2100 than in 2015 (Figure 4e). This increase was greater in Japan SSP5 than in Japan SSP1.

When the LULC change was investigated at a local scale, the difference in the spatial distribution of each LULC type between the population distribution changed. In Metropolitan Tokyo, where the total population was predicted to increase





until 2040 in Japan SSP1 and 2055 in Japan SSP5, the building area was predicted to increase. Simultaneously, paddy fields, other agricultural lands, and forests surrounding the densely inhabited district were predicted to decline. However, in other areas, where the population was predicted to decrease, the building areas were also predicted to decrease. In these areas, the wasteland was predicted to expand in both Japan SSP1 and Japan SSP5, while paddy fields, other agricultural lands, and forest were predicted to remain constant.

**5.4 Notes for using LULC data**

The following points should be considered when using the LULC data:

✓ Changes in building areas and wastelands are likely to be overestimated compared to actual LULC changes. In the methodology of this study, building areas were predicted to decrease and wastelands were predicted to increase in response to a decrease in population. In reality of a given area, vacant houses are unoccupied for several years, the

number of vacant houses increases, and eventually, the area transforms into a vacant area. Then, it takes several years for a vacant area to become a natural wasteland. However, we did not set a lag of years for the transition between the building area to the wasteland for these data.

✓ Since only direct population change is the driving factor for LULC change, changes in rice fields, other agricultural lands, and forests are likely to be underestimated compared to actual LULC changes. Only in the case of population

growth do rice fields, other agricultural lands, and forests decrease. Many regions are expected to undergo a decline in population, except in Tokyo prefecture. However, this effect was not considered in this study (e.g., increasing abandoned farmlands).

✓ We assumed that the areas of roads, railways, others, rivers and lakes, seashores, sea areas, and golf courses would not change from the current situation to 2100. However, there are roads and railways under construction or planned. Golf

courses are already in decline. Additionally, the area of rivers and lakes, and oceans would remain largely unchanged.

**6 Future challenges**

Possibilities for improvement of the common socio-economic scenarios are discussed below. First, it is necessary to verify the extent to which the narrative components of socio-economic scenarios (particularly Japan SSP1 and Japan SSP5) used here are consistent with the new Japanese policy. In October 2020, Yoshihide Suga, then Japanese prime minister,

announced a new Japanese policy which aims for the realization of a carbon-neutral, decarbonized society by 2050 (METI, 2020). The Japanese government aims to establish that carbon neutrality will lead to future corporate profits, creating a positive cycle of economic growth and environmental protection. The policy was set to meet efforts to limit a global-mean temperature rise of 1.5 °C under the 2015 Paris Agreement. Rogelj et al. (2018) clarified that in restricting global warming in the year 2100 to less than 1.5°C under assumptions of technological and resource development from global SSPs, it is

necessary for limiting end-of-century radiative forcing to 1.9 W m$^{-2}$. For ensuring the limit is not exceeded, the government





is faced with selecting pathways regarding decarbonization which include carbon capture and storage, carbon-dioxide removal (e.g. Bioenergy with Carbon Capture and Sequestration or large-scale afforestation), changes in land-use for $CO_2$ mitigation (e.g. restrictions on land development), energy system transformations, setting the facility of achieving deep mitigation, and investment in decarbonizing residual emissions from agriculture to reduce the emission intensity of food

production.

Second, when recent climate scenarios based on CMIP6 are used, LULC should be data consistent with the CMIP6 experimental setting. CMIP6 is being addressed multi-model climate projections based on alternative scenarios that are directly relevant to societal concerns regarding climate change mitigation, adaptation, or impacts. These climate projections were driven by a new set of emissions and LULC scenarios (Riahi et al., 2016) produced with integrated assessment models

(IAM) based on new future pathways of societal development, the SSPs, and were related to the RCPs. A new set of historical data based on the History of the Global Environment database, and multiple alternative scenarios of the future (2015–2100) from IAM teams, were provided as input for these models (Hurtt et al., 2020). These data have a spatial resolution of 0.25 degrees (~ 28km) on a global scale. The challenge is to achieve consistency between the global dataset and current national-scale database with different LULC classification types. Wu et al. (2020) developed a land use allocation

model for Japan with a spatial resolution of 0.05 degrees (～5km). The model is downscaled in its global version (Hasegawa et al., 2017) which is employed as the dataset in the SSP3-0.7 scenario. However, the LULC classification type remains that of the global dataset.

Third, to respond to requests from within the S-18 project, there are requirements to formulate the quantitate components that assume more detailed social and economic conditions. In addition to future projection of population, there are several

requirements. These include population projection by industry for assessing the impact of fisheries in theme 2 of the S-18 project and quality of Life sectors in theme 4, and projection of household for assessing local land use/urban environment in theme 4. Several requirements for LULC types are additions of greenery areas for assessing local land use/urban environment in theme 4 and abandoned farmland type for assessing economic impact on natural disasters in theme 5. The group assessing transportation system in theme 4 of the S-18 project requires future projection of changes in both roads and

railways. Projection methods for population by industry (Matsui et al., 2019), projection of household (Etoh and Onishi, 2018; NISSP 2019), changes in both roads and railways (Etoh and Onishi, 2018), abandoned farmland (Kobayashi et al., 2020) already exist. However, the projections in these previous studies did not take into account the Japan SSPs. The challenge is to incorporate consistency with the Japan SSPs regarding these methodologies.

## 7 Concluding remarks

The framework in the S-18 project aims to facilitate a wide range of nationally unified projections of climate change impacts and evaluation of adaptation options across multiple sectors. Setting of the common socio-economic scenarios for Japan is one element of the framework that also has both qualitative narratives and quantitate components over the 21st



century. The Japan SSP1, Japan SSP5, and status quo scenarios were designated as qualitative narratives components of socioeconomic scenarios. This study prepared a dataset of population and land use under these scenarios as the quantitate

components.

Compounding the climate scenario with the three socio-economic scenarios, which are characterizations of societal futures, will allow for analyses of future impacts and adaptation assessments that account for both climate and societal change in a coherent manner. In this paper, four combinations of two RCPs and the socio-economic scenarios were set. Throughout these sets, the outcome so far in the S-18 project to be generated from the five themes is expected to cover more than 60 impact

areas. Of these, more than 70% will be the analyses that consider both climate and socio-economic changes. Although we have future challenge of improvement of the socio-economic scenarios to adapting to both actual social conditions and government policies, and meeting the requirements for projection of multiple sectors, this approach achieved a successful outcome in the establishment of a common impact assessment framework to allow for comparisons across sectors at national and local scale. These outcomes will contribute as one element of the next impact assessments report for reviewing on both a

national and local adaptation plan. Moreover, these sets in Japan can help further environmental assessments in the future beyond the S-18 project.

The S-18 research project will summarize the results from all themes under socio-economic scenarios which were provided in this study. The S-18 project will have a second national assessment with wider areas. It aimed to improve the socio-economic scenarios through the future challenges as previously described. Ultimately, the success of the S-18 project lies in

the policy contribution to the adaptation planning at both local-scale and national-scale in Japan, and also in international contribution of new science and policy questions.

**Appendix A**

Table A1 summarizes previous studies of population projection. Some have no spatial distribution and others are single-year data. NIES (2021) delivers the population by the municipality and tertiary mesh distribution for each 5-year age group

under the Japan SSPs between 2015–2100. This is population projection only under the Japan SSPs. Table A2 summarizes previous studies of LULC projection. There are no studies which have developed the LULC projections associated with Japan SSPs.

**Author contribution**

The conceptualization of the research was developed by SY and NM. The scenarios including components of both

qualitative narratives and quantitative data were selected by SY, KT, KM and NM. The methodology of LULC projection was developed, and the LULC projection calculated by KT and WW. The tables and figures were produced by SY. The original draft of the paper was written by SY, with edits, suggestions, and revisions provided by KT, WW, KM and NM.



**Competing interests**

The authors declare that they have no conflict of interest.

**Code and data availability**

Code and data of land use scenario calculation in Yoshikawa et al. (2022, http://dx.doi.org/10.5281/zenodo.7096365) are available at Zenodo.

**Acknowledgements**

The authors would like to express our gratitude to Dr. Kei Gomi (Fukushima Regional Collaborative Research Center, National Institute for Environmental Studies) and Dr. Yuko Kanamori (Social Systems Division, National Institute for Environmental Studies) to provide population projection under Japan SSP1-5 through A-PLAT website (https://adaptation-platform.nies.go.jp/socioeconomic/population.html). This research was performed by the Environment Research and Technology Development Fund (JPMEERF20S11801) of the Environmental Restoration and Conservation Agency of Japan.

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



**Figures and Table**

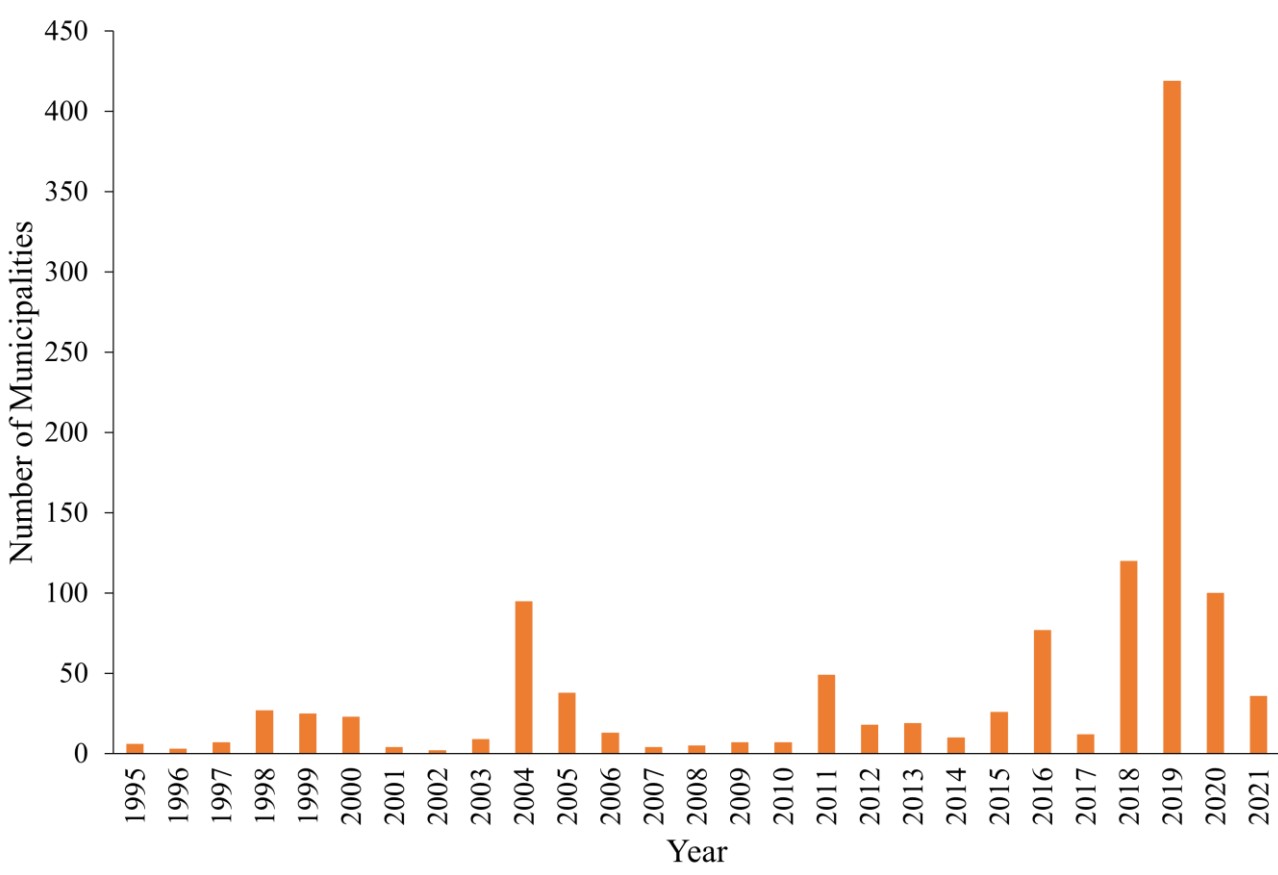


**Figure 1: Changes in number of municipalities where the Disaster Relief Act was applied due to storm and flood damage during 1995–2020. The main purpose of this Act is to protect victims of disaster and maintain social order by mandating that the government provide disaster relief. The number of municipalities was counted from reports of Disaster Management, Cabinet Office (https://www.bousai.go.jp/oyakudachi/info_saigaikyujo.html).**





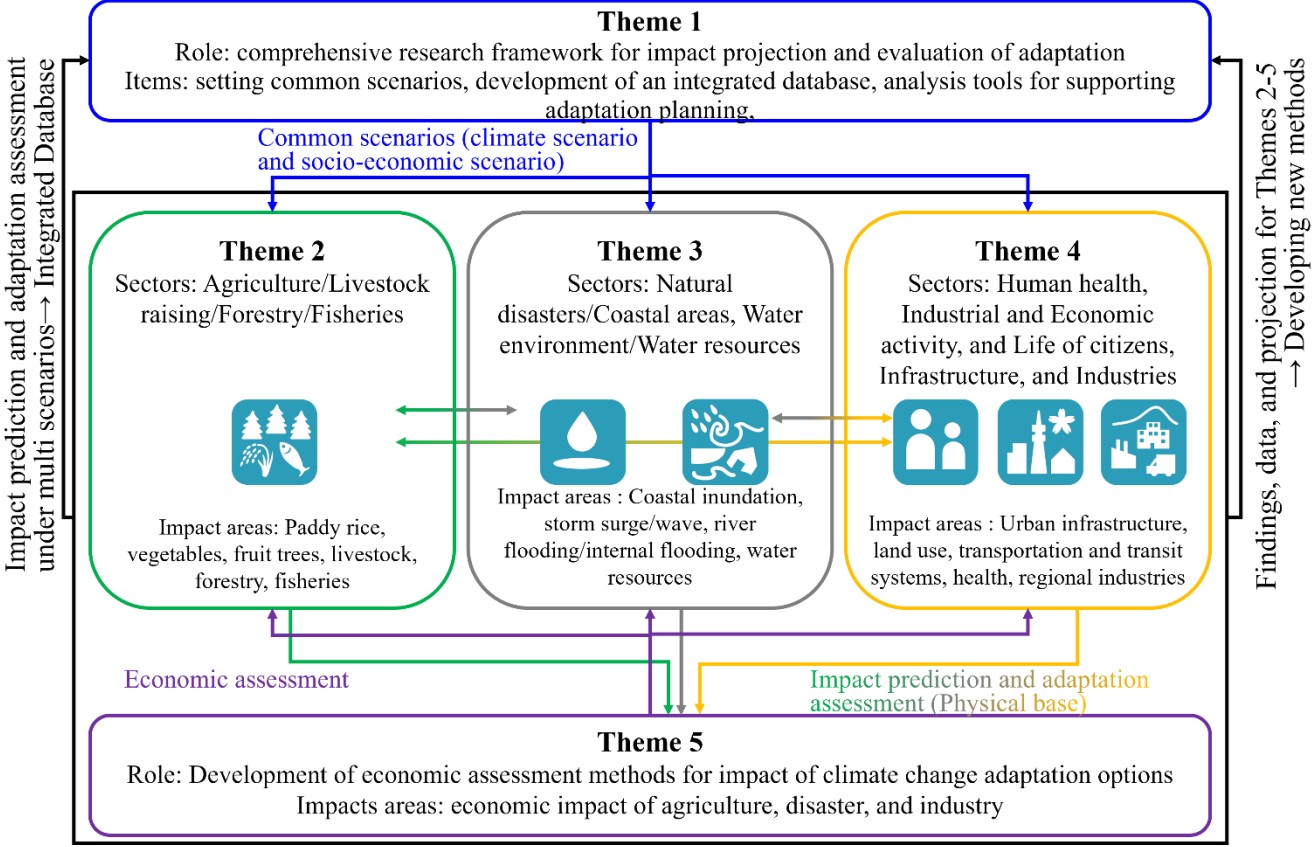

**Figure 2: Structure of S-18 research project. (Icon image source: Climate Change Adaptation Information Platform https://adaptation-platform.nies.go.jp/en/about/illustration.html)**




**Figure 3: Population projection from 2015 to 2100 under Japan SSP1 and Japan SSP5: a) changes in total population (millions); b) changes in population distribution between 2015 and 2100, expressed as age cohorts.**





**Figure 4: Predicted total areas of five LULC types from 2015 to 2100: a) paddy field; b) other agricultural land; c) forest; d) building area; and e) wasteland under Japan SSP1 and Japan SSP5, (1,000 km²).**






**Table 1: Summary of investigation results of socio-economic factors which S-18 researchers need.**

| Socio-economic factor | Total number of sub-themes required | Details of distribution | Sector | Impact areas | Temporal resolution | Spatial resolution |
|---|---|---|---|---|---|---|
| Land-use | 15 | Crop types | Water environment/water resources | River discharge | 5-year | Territory Units, municipal and prefecture scale |
| | | Paddy fields | Agriculture, forests/forestry, fisheries | Paddy rice yield | | |
| | | Tomato planted areas | Agriculture, forests/forestry, fisheries | Tomato leaf tip wilt incidence | | |
| | | Soybean planted areas | Agriculture, forests/forestry, fisheries | Soybean yield | | |
| | | Forest areas, croplands, urban areas | Agriculture, forests/forestry, fisheries | Height growth of cedar tree | | |
| | | Forest areas | Agriculture, forests/forestry, fisheries | Net cedar forest production | | |
| | | Forest type (Evergreen or Deciduous) | Agriculture, forests/forestry, fisheries | Net cedar forest production | | |
| | | Building areas | Life of citizens, infrastructure, and industries | Land use planning and regulation | | |
| | | Paddy Fields, other crops, Building areas, golf, traffic areas | Agriculture, forests/forestry, fisheries | Landslides in mountainous areas | | |
| | | | | Environmental response and vulnerability of forest tree species | | |
| | | | Natural disasters/coastal areas | Sea level rise in coastal areas | | |
| | | | | River and inland flooding | | |
| | | | Life of citizens, infrastructure, and | Building recycling systems | | |





| | | | | industries | | |
|---|---|---|---|---|---|---|
| | | Changes in croplands | Water environment/water resources | Available water for agricultural use | | |
| | | Residence area and greenery areas (e.g.,. Park, roadside trees) | Life of citizens, infrastructure, and industries | Vacant land within urban areas | | |
| Population | 9 | Population distribution by age group | Agriculture, forests/forestry, fisheries | Landslides in mountainous areas | 5-year | Territory Units, municipal and prefecture scale |
| | | | | Haul of fishes | 1-year | |
| | | | Natural disasters/coastal areas | Sea level rise in coastal areas | | |
| | | | | Sea level at storm surge | | |
| | | | | River and inland flooding | | |
| | | | Life of citizens, infrastructure, and industries | Building recycling systems | | |
| | | | | Residential population allocation | | |
| | | Industrial and employment population | Agriculture, forests/forestry, fisheries | Haul of fishes | | |
| | | Households distribution | Life of citizens, infrastructure, and industries | Residential population allocation | | |
| Asset distribution | 2 | Income distribution | Natural disasters/coastal areas | Sea level rise in coastal areas | | |
| | | Asset distribution | Natural disasters/coastal areas | Sea level at storm surge | | |







**Table 2: Land-use and land cover (LULC) types and descriptions.**

| LULC types | Descriptions |
|---|---|
| (1) paddy field | Irrigated rice field, wet fields, dry fields, swamp fields, lotus fields |
| (2) other agricultural land | Cultivated cropland such as wheat, upland rice, vegetables, grassland, lawn, apples, pears, peaches, grapes, tea, *Paulownia*, wax tree, *Broussonetia*, and *Trachycarpus* |
| (3) forest | Densely populated with perennial vegetation |
| (4) wasteland | Bamboo grass, wastelands, cliffs, rocks, perennial snow, wetlands, mined lands, etc. where the former land-use data is wasteland |
| (5) building area | Residential or urban area where buildings are densely built up |
| (6) roads | Highway, national road, prefectural road, municipal road |
| (7) railways | Railroads, railway shunting yard |
| (8) others | Athletic fields, airports, racetracks, baseball fields, schools, harbor areas, and vacant lots on land that has been built over with human engineering |
| (9) rivers and lakes | Artificial lakes, natural lakes, ponds, fishponds, fishponds; and river areas and the banks of rivers |
| (10) seashore | Area of sand, rubble, and rock bordering the sea |
| (11) sea area | Tide rock, mudflat, lagoons, and sea berth are included as sea |
| (12) golf course | Fairways and rough in the clustered part of the golf course; and areas outside the rough and up to the forest |






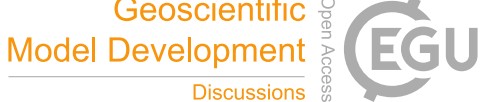

**Table A1: Summary of previous studies in population projection.**

| Reference | Projection Factor | Year | Spatial Resolution | Other | Consistency with Japan SSP |
|---|---|---|---|---|---|
| Hanasaki et al. 2012; 2014 | Population by Each Five-year age group | 2005-2100 Each 5-year | Japan MESH3 Boundaries (~1km) | - | Not Applicable |
| NIPSSP 2017 | Population by Each Five-year age group | 2020-2050 each 5-year | 500m | Households by prefecture level | Not Applicable |
| Etoh and Onishi 2018 | Population by Each Five-year age group Households | 2015-2100 Each 5-year | 109 water system | Households | Not Applicable |
| Sugiyama et al. 2019 | Total population | 2020-2050 Each 5-year | - | - | Not Applicable |
| Matsui et al. 2019 | Residential population and working population distribution in primary and industries | 2050 | 1-km | - | Not Applicable |
| Chen et al., 2020 | Total population | 2015-2100 Each 5-year | - | - | Applicable |
| NIES 2021 | Population by Each Five-year age group | 2015-2100 Each 5-year | Japan MESH3 Boundaries (~1km) | - | Applicable |





 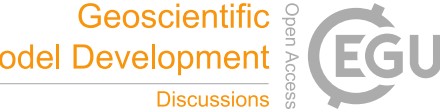

**Table A2: Summary of previous studies in LULC projection.**

| Reference | Projection Factor | Year | Basic year | Basic map | Spatial Resolution | Consistency with Japan SSP |
|---|---|---|---|---|---|---|
| Hanasaki et al. 2012; 2014 | Paddy, Other agriculture, Forest, Wasteland, Building, Roads and Rail, Other, Rivers, Seashore, Sea, Golf | 2005-2100 Each 5-year | 2006 | MLIT (2006) | Japan MESH3 Boundaries (~1km) | Not Applicable |
| Eto and Onishi 2018 | Building (House, Hotel, Shop etc.), Roads, Railways, | 2015-2100 Each 5-year | 2015 | Arc GIS Data Collection 2015 | Japan MESH3 Boundaries (~1km) | Not Applicable |
| Ohashi et al. 2019 | Paddy fields, Forest, Wasteland, Build-up area, Other agricultural land, Other artificial land | 2015-2050 | 1985-2010 | MLIT (1976, 1987, 1991, 1997, 2006, 2009, and 2014) | Japan MESH3 Boundaries (~1km) | Not Applicable |
| Shoyama et al. 2019 | Residential area, paddy field, cropland, other agricultural land, bush, grassland, and other vegetation, natural forest, secondary forest, plantation forest, and others | 2050 | 1987, 1998, 2014 | Ogawa et al. (2013) | 1km | Not Applicable |
| Wu et al. 2020 | Rice irrigated, Wheat irrigated, Other coarse grains irrigated, Oil crops irrigated, Sugar crops irrigated, Rice rainfed, Wheat rainfed, Other coarse grains rainfed, Oil crops rainfed, Sugar crops rainfed, Other crops, Bioenergy crops, Afforestation, Settlement, Ice and water, Other land-use, Forest, Pasture, Other natural vegetation | 2010-2100 | 2006 | MLIT (2006) | 0.05 degree | Not Applicable |

Reference: Ogawa, M., Takenaka, A., Kadoya, T., Ishihama, F., Yamano, H., Akasaka, M.: Land-use classification and mapping at a whole scale of Japan based on a national vegetation map. Jpn J Conserv Ecol 18:69–76, 2013, in Japanease.