# Peer review of "Development of common socio-economic scenarios for climate change impact assessments in Japan"

_Geoscientific Model Development, 2022_

## Referee Comment (RC1)

This study discusses and develops socio-economic scenarios for Japan that are consistent with Shared socio-economic pathways (SSPs). As the authors mentioned, the scenarios developed in the S-18 project and this study will be useful for detailed assessment of climate change impact in Japan. Besides, this paper carefully explains framework of the project, related studies, and so on, and easy to read. Thus, it is potentially an important study.

On the other hand, regarding methodology, there are many parts revisions are needed. Major comments are listed below:

Population scenarios (Section 4)
- The authors just use existing population scenario data (i.e., Japan SSP1 and Japan SS5 of NIES, 2021). Therefore, in terms of population projection, there is no novelty. Because of the reason, I am not sure if Section 4 is really needed.
- A more detailed explanation on how Japan SSP1 and Japan SSP5 are projected is needed to understand these scenarios.
- In my understanding, Japan SSP1 and Japan SSP5 provides population projection with adjustment based on their own scenarios. To clarify the influence of these scenarios on future population, it is desirable to additionally compare these scenarios with population projection without adjustment based on scenario (i.e., population under business-as-usual scenario).

LULC scenarios (Section 5)
- A building area projection model is proposed in Section 5.2. However, it is unclear how reliable and accurate this model is. It is needed to examine if the model accurately explains the relationship between accrual population and building area.
- Related to the previous comment, when projecting LULC, it is typical to estimate LULC transition matrix (e.g., a matrix whose (i,j)-th element denotes the probability that the i-th LULC changes to j-th LUCL) from past data, and use it for future projection. Such a transition matrix-based projection is likely to be more accurate and interpretable than the model assumed in this study. It is needed to explain why the authors rely on the simple model assuming linear relationship between population and building area.

Other minor comments are as follows:
- Based on the abstract, the contribution of this paper is that "we established common socio-economic scenarios designated as Japan SSP1, Japan SSP5, and status quo". However, it is an overstatement because the authors created LULC scenarios only (regarding population scenario, existing scenarios are used).
- Figure 1: The period in the graph is 1995-2021 but the caption says "during 1995-2020"

- Line 270: areaa -> area
- Line 269: "i" seems representing mesh code rather than the number of mesh units. Please check.
- Figure 3: In the current color coding, urban areas seem uniformly orange, and it is hard to visually distinguish difference in population size in each area. A better color coding is needed to clarify population difference in each area.
- The titles of Section 5.2 "Creation of future scenarios for LULC distribution based on population data" and Section 5.3 "Future scenarios for LULC distribution based on population data" are too similar.
- In the draft, NISSP, NIPSSP, and NIPSSR, which seem to have the same meaning, appear. Please unify them for consistency.

---

## Author Comment (AC1)

**Dear Reviewer 1:**

Thank you very much for taking the time to review our manuscript. All your comments were tremendously helpful for the improvement of the paper. We have responded to all the comments below. We believe that all your concerns have now been addressed.

On behalf of all authors,
Sayaka Yoshikawa

This study discusses and develops socio-economic scenarios for Japan that are consistent with Shared socio-economic pathways (SSPs). As the authors mentioned, the scenarios developed in the S-18 project and this study will be useful for detailed assessment of climate change impact in Japan. Besides, this paper carefully explains framework of the project, related studies, and so on, and easy to read. Thus, it is potentially an important study. On the other hand, regarding methodology, there are many parts revisions are needed. Major comments are listed below:

> Thank you for your positive evaluation of our manuscript. We completely agree with your comments; we have responded to them individually.

Population scenarios (Section 4)
- The authors just use existing population scenario data (i.e., Japan SSP1 and Japan SS5of NIES, 2021). Therefore, in terms of population projection, there is no novelty. Because of the reason, I am not sure if Section 4 is really needed.

> As described in the manuscript, this is indeed a dataset that already exists. However, we would like to leave Section 4 to introduce them, because this is the first study to apply the population projection scenarios as a part of a common socioeconomic scenario for the integrated impact assessment to Japan. In addition, we do not see an issue with the description here, since GMD states that datasets in "model experiment description papers," can include data that has already been published elsewhere. However, as pointed out below, there are some missing details regarding the differences between Japan SSP1 and Japan SSP5 and their comparison with BAU. To clarify these points, we have revised Section 4 (Lines 220-225 and 232-248 in the revised manuscript).

- A more detailed explanation on how Japan SSP1 and Japan SSP5 are projected is needed to understand these scenarios.

Thank you for pointing this out to us. We have added the following detailed explanation of the two scenarios in the manuscript (Lines 220-225 in the revised manuscript).

"In the Japan SSP1, the population decline is relatively relaxed, with a high birth rate of 18% (total population 73 million people) by 2100, because childcare environment investments for education towards creative human resources development accelerate the birth rates. Only medium rates exist for mortality rates and migration rates. In contrast, the Japan SSP5 assumed increasing net immigration of non-Japanese origin, which will be of 250,000 people by 2035 (total population 79 million people by 2100) because labor markets are gradually opened and international mobility is increased. Medium birth rates and mortality rates are set."

- In my understanding, Japan SSP1 and Japan SSP5 provides population projection with adjustment based on their own scenarios. To clarify the influence of these scenarios on future population, it is desirable to additionally compare these scenarios with population projection without adjustment based on scenario (i.e., population under business-as-usual scenario).

Thank you for pointing this out to us. We agree that a comparison with "Population projections without scenario-based adjustments" is important to understand what kind of scenario we used. Therefore, we have added the following description in the manuscript (Lines 232-248 in the revised manuscript).

"The population projections used (Japan SSP1 and Japan SSP5) were compared to those without scenario-based adjustments. There are two types of the projections without the adjustments. The first is a national population projection under SSP1 and SSP5 developed by the International Institute for Applied Systems Analysis and the National Center for Atmospheric Research (Global SSP1 and Global SSP5 in Figure 4; Riahi et al., 2017). The projection using multi-dimensional mathematical demography is based on alternative assumptions on future fertility, mortality, migration, and educational transitions that correspond to the SSP storylines (Samir and Lutz, 2017). Figure 4 shows a comparison of the total population change under Japan SSP1 and Japan SSP5 with global SSP1 and global SSP5. The

total population of Japan SSP1 is 5.3% lower than global SSP1, while the population of Japan SSP5 is 24.6% lower than global SSP5 in the year 2100 (described in Chen et al., 2020). Global SSP1 and Global SSP5 assumed medium and high fertility, respectively, and low mortality. This is not appropriate for Japan, where there is a declining birthrate and an aging population. Therefore, Japan SSP1 and Japan SSP5 assumed high and medium fertility, respectively, and medium mortality. The second type of the projections without the adjustments is a medium population projection in NIPSSR (2017), that is consistent with typical patterns of historical experience in Japan. In it, the total population of Japan SSP1 is 22% higher than in the medium projection, while the population of Japan SSP5 is 25% higher in the year 2100 (Figure 4). Although the NIPSSR's medium projection assumed medium fertility and mortality, Japan SSP1 assumed a high fertility. Japan SSP5 assumed a medium level of both fertility and mortality. This is the reason why the population decline under both Japan SSP1 and Japan SSP5 is suppressed compared to the NIPSSR's medium projection."

[Figure]

Figure 4: Comparison of total population change under Japan SSP1 and Japan SSP5 with national population projection under Global SSP1 and Global SSP5 (Rihai et al., 2017) and medium projection in NIPSSR (2017).

LULC scenarios (Section 5)

- A building area projection model is proposed in Section 5.2. However, it is unclear how reliable and accurate this model is. It is needed to examine if the model accurately explains the relationship between accrual population and building area.

> Thank you for pointing this out to us. We have added the following text in section 5.4 of the revised manuscript to address this (Lines 350-364 in the revised manuscript).
>
> "LULC change is a complex process that is affected by diverse socio-economic factors. There are also LULC projection models describing the complex process considering multiple socio-economic factors. However, a major challenge for utilizing the models in a long-term LULC projection is that we also need to prepare acceptable long-term future scenarios for all the explanatory variables, that could be an additional source of uncertainty and might increase difficulty in communicating features of the projected LULC scenarios with users. For the period with population increase, we expect that a building area also increased more or less. However, for the future period with rapid decline in population, we do not have enough empirical evidence for developing a reliable LULC model. Although in fact Japan is one of the countries with the largest population decline in the world, there are almost no examples of the relationship between socio-economic changes associated with rapid declining population recent years and land use change. To avoid increasing the uncertainty of future LULC projections by increasing the number of assumptions, we decided to adopt a simple rule-based LULC estimation method with population change as the driving factor. Demographic change was used as the major driver of LULC change also in the previous studies on LULC projections in Japan, (e.g., Hanasaki et al., 2012; 2014; Etoh and Onishi 2018; Ohashi et al., 2019; Shoyama et al., 2019). We thus adopted the approach similar to the previous studies, although we also recognize the improved projection of LULC is a major challenge for the future study."

- Related to the previous comment, when projecting LULC, it is typical to estimate LULC transition matrix (e.g., a matrix whose (i,j)-th element denotes the probability that the i-th LULC changes to j-th LUCL) from past data, and use it for future projection. Such a transition matrix-based projection is likely to be more accurate and interpretable than the

model assumed in this study. It is needed to explain why the authors rely on the simple model assuming linear relationship between population and building area.

While it is highly challenging to make precise projections of land use change 80 years from now, it is needed to prepare land use assumptions usable for multiple sectors in the S-18 project. The reason why we followed the simple approach is explained in our response to the above comment. The reviewer pointed out that the transition matrix or its derivatives may be more appropriate, and we agree with that statement regarding short-term land use change. However, we did not use it because its compatibility with the depiction of long-term scenarios up to 2100 is not certain and should be studied further.

Other minor comments are as follows:

- Based on the abstract, the contribution of this paper is that "we established common socio-economic scenarios designated as Japan SSP1, Japan SSP5, and status quo". However, it is an overstatement because the authors created LULC scenarios only (regarding population scenario, existing scenarios are used).

Thank you for pointing this out. We have revised the contribution part of the abstract.

- Figure 1: The period in the graph is 1995-2021 but the caption says "during 1995-2020"

This was a mistake in the figure caption. We have modified it.

- Line 270: areaa -> area

Thank you for pointing this out. We have corrected it.

- Line 269: "i" seems representing mesh code rather than the number of mesh units. Please check.

As you pointed out, the "i" is a mesh code. We have modified it.

- Figure 3: In the current color coding, urban areas seem uniformly orange, and it is hard to visually distinguish difference in population size in each area. A better color coding is needed to clarify population difference in each area.

We have modified Figure 3b) to improve its visibility.

- The titles of Section 5.2 "Creation of future scenarios for LULC distribution based on population data" and Section 5.3 "Future scenarios for LULC distribution based on population data" are too similar.

We have changed the heading of section 5.3 to "Future scenarios for LULC."

- In the draft, NISSP, NIPSSP, and NIPSSR, which seem to have the same meaning, appear. Please unify them for consistency.

Thank you for pointing out that NISSP and NIPSSP were misprints. The correct name is NIPSSR. We have corrected all the errors in the manuscript.

---

## Author Comment (AC2)

**Dear Reviewer 2:**

Thank you very much for taking the time to review our manuscript. Your comments were tremendously helpful in improving the paper. We have responded to all the comments below. We believe that all your concerns have now been addressed.

On behalf of all authors,
Sayaka Yoshikawa

The authors of this manuscript are trying to develop common socioeconomic scenarios with some details (qualitative and quantitative) for Japan. However, as I read the manuscript, most of the information and data were taken from literature and what the authors did was estimation of LULC (and the methodology is taken from the literature (some updates but not sure about the improvement). Therefore, no scientific contributions were found.

> Thank you for your critique which gave us a chance to consider the scope and feature of our paper again.
>
> Projections of climate change impacts on a national level need to consider not only climatic hazard but also exposure and vulnerability of the socio-economic system. However, there is no established method for estimating them because the socio-economic change at a national or even regional level is determined by a variety of context-dependent factors spanning a long period. Therefore, we tried to develop socio-economic scenarios combining the best available data and methods for population and LULC projections based on the Japan SSPs. These scenario data have been used for the impact assessment in several fields. While the national and regional level assessment of impacts needs to be further developed, this paper presents an approach on how to integrate the data and methods, both existing and new, for the impact assessment of climate change to inform policy decisions. We think that this is the scientific contribution of our paper. In addition, the scope and contents of our manuscript meet the GMD requirements for the category "model experiment description papers."

In addition, the authors focused too much on their project, which is unnecessary information for scientific papers (except for the acknowledgment). Therefore, the authors must delete such information.

The scenario set-up for climate change impact and adaptation assessment is an essential basis of the S-18 research project. Therefore, we need to explain the target, objectives and structure of the S-18 research project to inform readers of the relationship between the socio-economic scenarios and the impacts and adaptation assessment.